# NOVEL POSITIONAL ENCODINGS TO ENABLE TREE-STRUCTURED TRANSFORMERS

## ABSTRACT

With interest in program synthesis and similarly flavored problems rapidly increasing, neural models optimized for tree-domain problems are of great value. In the sequence domain, transformers can learn relationships across arbitrary pairs of positions with less bias than recurrent models. Under the intuition that a similar property would be beneficial in the tree domain, we propose a method to extend transformers to tree-structured inputs and/or outputs. Our approach abstracts transformer's default sinusoidal positional encodings, allowing us to substitute in a novel custom positional encoding scheme that represents node positions within a tree. We evaluated our model in tree-to-tree program translation and sequence-to-tree semantic parsing settings, achieving superior performance over the vanilla transformer model on several tasks.

## 1 INTRODUCTION

### 1.1 SEQUENCE MODELING

Neural networks have been successfully applied to an increasing range of tasks, including speech recognition and machine translation. These domains crucially depend on techniques for modeling streams of audio and text, represented as dynamically sized sequences of tokens. Researchers have historically handled such data primarily with recurrent techniques, which encode sequences into fixed-length representations. The sequence-to-sequence LSTM model (Sutskever et al. (2014)) is a particularly notable example in recent times.

Recurrent architectures have some disadvantages. From a generalization perspective, recurrent cells face the challenge of learning relationships between tokens many time steps apart. Attention mechanisms are now commonly employed to mitigate this issue, driving new state-of-the-art results in difficult tasks such as machine translation (Wu et al. (2016)). From an efficiency standpoint, recurrence does not lend itself to parallelism, often rendering recurrent models expensive to train. Recurrent models are also difficult to interpret, employing an obtuse series of neural layers between time steps that render relationships modeled within the data unclear.

The transformer (Vaswani et al. (2017)) is a stateless sequence-to-sequence architecture motivated by these issues, constructed by forgoing recurrence altogether in favor of extensive attention. This design allows information to flow over unbounded distances during training and inference, without the need for complex gates and gradient clipping. This type of long distance flow, driven by learned attention transforms over positional encoding, provides a powerful computational mechanism. Transformers also lend themselves to easier interpretation, as their attention layers can at least reveal information about learned relationships between elements of a sequence.

### 1.2 HIERARCHICAL MODELING

Recent work has begun to apply neural networks to programming tasks (Allamanis et al. (2018)). In recent years, programming language analysis techniques have begun to exploit statistical techniques commonly used on large natural language corpora (Hindle et al. (2012)). These can be used to identify idioms in software, enable searching for code clones, searching code by natural language, or even translating from one programming language to another.

Representing programs is an interesting challenge. One option is to view them as a one dimensional sequence of tokens and use techniques common in the natural language programming literature. However, these programs are intentionally endowed with hierarchical structure, even graph-like relations. Using purely sequence-oriented methods may result in losing valuable information about this structure.

Expanding past sequential modeling, a common approach is to pass information through neighbors in the graph, in a manner that is reminiscent of message passing in graphical models (Li et al. (2016)). To ensure that information can fully propagate across the graph, this message-passing must be applied multiple times, bounded by the diameter of the graph. While this allows us to exploit hierarchical structure, ideally we would like to do so while capturing the efficient information flow and other benefits of transformer models.

In this work, we generalize transformers to embed tree representations. Our work introduces novel positional encodings for tree-structured data. Using these encodings, we can apply transformers to tree-structured domains, allowing information to percolate fully across the graph in a single layer. This can potentially extend the transformer to settings ranging from natural language parse trees to program abstract syntax trees. We evaluate our tree-transformers on programming language translation (e.g., translating JavaScript to CoffeeScript) (Chen et al. (2018)) and semantic parsing (e.g. extracting a database query from a natural language request Dahl et al. (1994)) tasks, demonstrating improved performance over sequential transformers.

## 2 BAG-OF-POSITIONS INTERPRETATION

The order of a sequence is rich in information, and order-agnostic (bag-of-words) models are limited in power by their inability to use this information. The most common way to capture order is through recurrence; recurrent models inherently consider the order of an input sequence by processing its elements sequentially. In the absence of recurrence, we require additional information about the input sequence's order in some other form. Transformer models provide this additional information in the form of positional encodings. Each position in the input sequence is associated with a vector, which is added to the embedding of the token at that position. This allows the transformer to learn positional relationships between tokens as well as relationships between token embedding and positional encoding space.

While adding in positional encodings addresses the power limitations of bag-of-words representations, it fundamentally does so by upgrading the bag of words to a bag of annotated words. Indeed, the transformer's core attention mechanism is order-agnostic, treating keys as a bag. The calculations performed on any given element of a sequence are entirely independent of the order of the rest of that sequence in that layer; this leaves most of the work of exploiting positional information to the positional encodings and autoregressive property.

Now, a bag of words annotated with positions can be equivalently thought of as a bag of positions annotated with words. From this perspective, we see that it is not at all necessary that our input "sequence" of positions have any direct correspondence with the sequence of associated "indices," i.e. an evenly distributed number line. The original transformer's positional encodings do form this correspondence for the purposes of sequence modeling, but we can consider more arbitrary positional encodings to represent more arbitrary structures within positional space, as long as the relationships between points in positional space have some useful semantic meaning. In particular, we can use this idea to extend the transformer to the tree domain.

## 3 TREE POSITIONAL ENCODINGS

Now we construct our positional encoding scheme for trees. The transformer's original positional encodings has two key properties we would ideally like to preserve. First, every position has a unique positional encoding, so that attention to any given position can be sharply defined. Second, any relationship between two positions can be modeled by an affine transform between their positional encodings. This allows the transformer to efficiently learn relationships between positions within its embedding layers. In the context of sequences, the relationship between two positions is simply the distance $k$ that separates them. For trees though, the relation between two nodes is a *path*: a series

Figure 1: Examples of how to compute positional encodings for nodes in a regular tree. The sequence of branch choices $b$ is used to determine a sequence of transforms $D_{b_1}, D_{b_2}, \ldots$ to apply to the root node's positional encoding. $U$ is complementarily defined such that applying it to any of these nodes results in that node's parent (e.g. $r = Ux = U^2y = U^3z$). The specific transforms $D_i, U$ are defined in Equations 1 and 2.

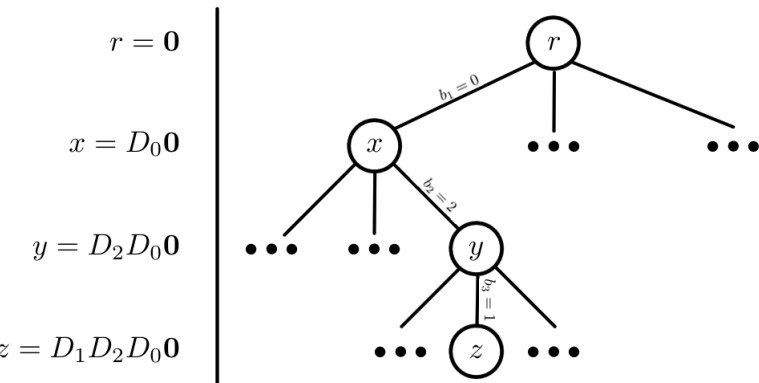

of steps up or down along tree branches. Our positional encoding scheme should try to associate each such path with a unique linear transform in a consistent way.

From a given node in an $n$-ary tree, there are $(n + 1)$ potential length-1 paths: a branch down to each of $n$ children, and a branch up to the parent. We will denote the branches down to children as $D_{1,\ldots,n}$ and the branch up to parent as $U$. We can then denote any longer path as a composition of these $D$'s and $U$'s, which act as operators. For example, if we wish to denote a node $x$'s grandparent's second child, we can write $D_2U^2x$. Every path can be broken down into a composition of these $(n + 1)$ operators, so we need only focus on their relationships. We want to associate each operator $U, D_{1,\ldots,n}$ with a unique affine transform; for convenience, we will also refer to their affine transforms as $U, D_{1,\ldots,n}$ respectively, and to $x$'s positional encoding as simply $x$.

The fundamental relationship between these operators is that traveling up a branch negates traveling down any branch. Our constraint then is:

$$UD_i = I \; \forall i \in \{1, \ldots, n\}$$

The positional encoding scheme we propose adheres to this constraint for all trees up to a specified depth, and still works well in practice for even deeper trees. We will first explain the parameter-free form of our positional encoding scheme for simplicity. Our scheme takes two hyperparameters: $n$, the degree of our tree, assumed to be regular; and $k$, the maximum tree depth for which our constraint is preserved. Each positional encoding has dimension $n \cdot k$, and each transform $U, D_{1,\ldots,n}$ preserves this dimensionality. We assign the root a zero vector, and define every other node by its path from the root vector. We denote this path as $b_1, \ldots, b_L$, where $b_i$ is the branch choice at the $i$th layer and $L$ is the layer at which the node resides. Then, for any node $x$, we compute its positional encoding:

$$x = D_{b_L} D_{b_{L-1}} \ldots D_1 \mathbf{0}$$

Now we describe the affine transform of $D_i$. We represent a move down along $x$'s $i$th branch by concatenating a one-hot $n$-vector with hot bit $i$ ($e_i^n$) to the left side of x, and truncating x on the right to preserve dimensionality. We define $U$ complementarily. In other words,

$$D_i x = e_i^n; x[: -n] \tag{1}$$
$$U x = x[n :]; \mathbf{0}_n \tag{2}$$

where ; represents concatenation, and $[n :]$ and $[: -n]$ represent truncatation by $n$ elements on the left and right side, respectively (as per Python notation).

These $D, U$ satisfy our constraint whenever $L \leq k$. Note that for trees with depth greater than $k$, $UD_i$ is not necessarily the identity. Traveling down more than $k$ layers will cause this scheme to

Figure 2: Nearest neighbor heatmap of parameter-free tree encoding scheme. We number the nodes in the tree according to a breadth-first left-to-right traversal of a balanced binary tree: position 0 is the root, 1 is the first child of root, 2 is the second child of root, 3 is the first child of the first child of root, and so on. In each case, we consider the row position as a "query" and each column position as a potential "value". The attention score of solely the positional encoding after softmax is represented as a heatmap scaling from black (0.0) through red and yellow to white (1.0).

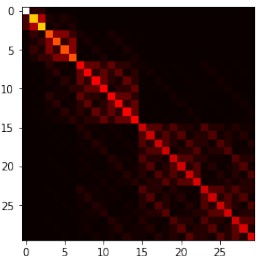

Figure 3: Nearest neighbor heatmaps of parameterized tree encodings with example values of $p$. As shown in Figure 2, many of the lower-level positions in the tree are quite similar in the absence of a decay factor. For example, position 5 (Root,$D_2$,$D_1$) is most similar to itself (score of 0.44), but quite similar to position 6 (Root,$D_2$,$D_2$) and position 3 (Root,$D_1$,$D_1$) with scores of 0.16. An appropriate level of decay allows each position to be uniquely identified as in (a); too much decay provides little additional information as in (b).

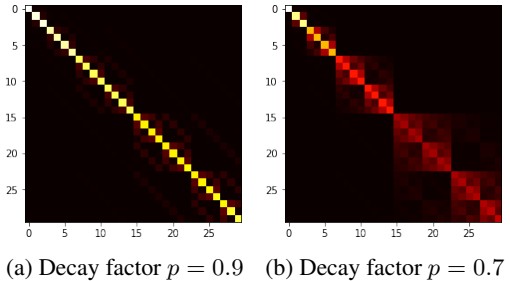

(a) Decay factor $p = 0.9$     (b) Decay factor $p = 0.7$

"forget" nodes more than $k$ layers up, which cannot be inverted. In practice, we make the simplifying assumption that this loss of information is insignificant for sufficiently large $k$.

The parameter-free positional encoding scheme as proposed so far, while fulfilling the uniqueness property and approximately the linear composition property, lacks richness. It is analogous to a simplified sequential positional encoding scheme that simply defines the positional encoding at index $i$ to be the number $i$. The transformer instead opted for a rich stack of sinusoids of different frequencies to attend to a much wider variety of relationships. In a similar vein, we propose adding a parametrizable component to diversify our encodings.

Our encoding consists of a sequence of one-hot chunks, each representing a different layer of the tree. One will note that we can weigh these one-hot chunks with any geometric series without disrupting the affine property:

$$x' = x \odot (\mathbf{1}_n; \mathbf{p}_n; \mathbf{p^2}_n; \dots)$$

$x'$ here meets the same properties as $x$. Here, $p$ is a parameter and $\mathbf{p}_n$ is a $n$-vector of $p$'s. As Figure 3 shows, different values for $p$ result in radically different attention biases. Analogous to the stack of sinusoidal encodings, we propose a stacking multiple tree encodings, each equipped with its own $p$ to be learned. To prevent the encodings' norms from exploding, we apply $\tanh$ to p to bound it between -1 and 1, and multply the encodings by a factor of $\sqrt{1 - p^2}$ to approximately normalize it. We then scale it further by a factor of $\sqrt{d_{model}/2}$ to achieve norms more similar to the original transformer's positional encoding scheme.

Figure 4: Common traversals and mixtures thereof can be represented as linear transforms. Using the position encoding described in this paper, finding the parent, left child, or right child of a given node can be represented as linear transforms $U$, $D_1$, and $D_2$. Complex traversals can be represented also as linear transforms by composing these operations. The attention heatmaps below demonstrate the similarity of tree positional encodings applied to different points in the tree when the "query" has been transformed before dot product with the value.

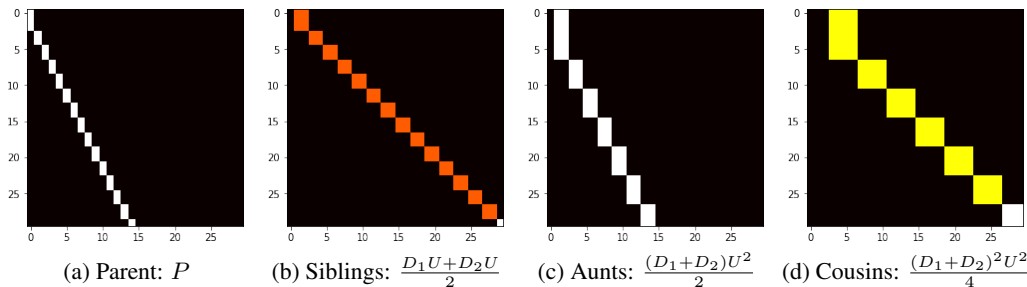

(a) Parent: $P$     (b) Siblings: $\frac{D_1U + D_2U}{2}$     (c) Aunts: $\frac{(D_1+D_2)U^2}{2}$     (d) Cousins: $\frac{(D_1+D_2)^2U^2}{4}$

## 4 DECODER

To accommodate a new positional encoding scheme, we need to slightly modify the decoder. The original transformer's decoder concatenates a start token to the beginning of the sequence without modifying the positional encodings. This results in misalignment between autoregressed outputs and positional encodings, e.g. the encoding for the second position is summed with the embedding of the first output. This is not an issue in the sequential case; the positional encodings are self-similar, so this "misalignment" is a linear transform away from the "correct" alignment. However, no traversal through a tree's nodes have this self-similarity property, so proper alignment here is critical.

We use a zero vector for the start token's positional encoding, and use the appropriate positional encoding for each autoregressed output. Our decoder must dynamically compute the new positional encoding whenever it produces a token. The decoder must keep track of the partial tree structure that it constructs, to correctly traverse to the next position based on history. In order to build this partial tree structure, the decoder must be aware of how many children each node must have. To this end, we construct our vocabularies such that each symbol is annotated with a number of children. When symbols have a varying number of children, they are added multiple times to the vocabulary, each with a different annotation. Given this information, the decoder can flexibly construct trees using any tree traversal algorithm, as long as it is applied consistently. In our experiments, we explored both depth-first and breadth-first traversals for decoding.

## 5 EXPERIMENTS AND RESULTS

For our evaluation, we consider both tree-to-tree and sequence-to-tree tasks. Both categories test our model's ability to decode tree-structured data; the sequence-to-tree task additionally tests our model's ability to translate between different positional encoding schemes. Our tree-to-tree evaulation centers around program translation tasks, while our sequence-to-tree evaluation focuses on semantic parsing.

As our model expects regular trees, we preprocess all tree data by converting trees to left-child-right-sibling representations, which are binary trees. This enforces $n = 2$ for our model. We use a maximum tree depth $k = 32$ for all experiments. [1] Unless listed otherwise, we performed all of our experiments with Adam (Kingma & Ba (2015)), a batch size of 128, a dropout rate of 0.1 (Srivastava et al. (2014)), and gradient clipping for norms above 10.0.

---

[1] Although binarizing trees may not always be necessary, both programming language and natural language trees of ten have constructs with unbounded numbers of children (e.g. statement blocks). For years, natural language parsing efforts have converted n-ary grammars into binary forms to enable efficient algorithms and estimation (Klein & Manning (2003)). We explored left-child-right-sibling representations primarily to be consistent with past work (Chen et al. (2018)); it would be interesting to measure the impact of alternate binarization strategies (or omitting binarization altogether) when using tree transformers.

|  | SYN-S | SYN-L |
|---|---|---|
| Tree-transformer, depth-first search | **99.98** | **98.77** |
| Tree-transformer, breadth-first search | 99.86 | 95.93 |
| Seq-transformer, parse trees | 99.69 | 94.43 |
| Seq-transformer, raw programs | **99.99** | **99.46** |
| Tree2tree LSTM (Chen et al. (2018)) | 99.76 | 97.50 |
| Seq2seq LSTM (Chen et al. (2018)) | 98.38 | 12.19 |

Table 1: Program accuracy data for synthetic tasks. The tree-transformer has clear advantages over the sequence-transformer for larger tasks, indicating that the custom positional encodings may be providing useful structural information.

## 5.1 TREE-TO-TREE: PROGRAM TRANSLATION

For tree-to-tree evaluation, we focused on three sets of program translation tasks from the literature to test our model against. The first set of tasks is For2Lam, a synthetic translation dataset between an invented imperative and functional language. The dataset is split into two tasks: one for small programs and one for large programs. The second set of tasks involves translating between generated CoffeeScript and JavaScript code. The data is similarly broken, here both by program length and vocabulary. More details about the data sets can be found at (Chen et al. (2018)). We report all results in terms of whole program accuracy.

### 5.1.1 SYNTHETIC TRANSLATION TASKS

For the synthetic translation tasks, we trained both our tree-transformers and classic sequence-transformers for baseline experimentation. We trained four models for each task: a sequence-transformer that operates on raw programs; a sequence-transformer that operates on parse tree representations; a tree-transformer with breadth-first traversal; and a tree-transformer with depth-first traversal. Both models were trained with four layers and $d_{model} = 256$. The sequence-transformer was trained with $d_{ff} = 1024$ and a positional encoding dimension that matched $d_{model}$, in line with the hyperparameters used in the original transformer. The tree-transformer, however, was given a larger positional encoding size of 2048 in exchange for a smaller $d_{ff}$ of 512. This was to emphasize the role of our tree positional encodings, which are inherently bulkier than the sequential positional encodings, while maintaining a similar parameter count.

The results for the synthetic tasks can be found in Table 1. Both forms of transformer, as well as all other baseline methods listed, get very close to solving the small program dataset. The results on the large program are of more interest: both tree-transformer models perform significantly better than the sequence-transformers, suggesting that the positional encodings help considerably for larger trees. The depth-first search variant outperforms breadth-first search in both cases. We conjecture that depth-first search may be a more favorable traversal method in general; it tends to construct more subtrees similar to each other earlier in the process. Interestingly, the tree-transformer over tree-structured data performed slightly worse than the sequence-transformer over raw program representations. The sequence-transformer may benefit from the overall short length of programs in these datasets.

### 5.1.2 COFFEESCRIPT-JAVASCRIPT TRANSLATION

Given the results on the synthetic tasks, we focused on training depth-first traversal tree-transformers for this task. The data is partitioned four ways, into two sets of vocabulary ('A' and 'B') and two categories of program length (short and long). We use the same hyperparameters as in the synthetic tasks, and once again compare our results with the tree2tree and seq2seq models. For memory-related reasons, a batch size of 64 was used instead for the tasks with longer program lengths.

The results for CoffeeScript-Javascript translation can be found in Table 2. The tree-transformer obtains state-of-the-art results on over half the datasets, while still producing competitive results on the other datasets. This results demonstrate that the advantages of the tree-transformer's design are more prominent with large data. While its performance tends to be weaker on the simpler short-sequence tasks, the tree-transformer gains up to 20 percentage point improvements over the state

|  | Tree-Tform | Tree2tree | Seq2seq |
|---|---|---|---|
| CJ-A-short | 96.51 | **99.57** | 92.73 |
| JC-A-short | **97.71** | 87.75 | 86.31 |
| CJ-B-short | 97.80 | **99.75** | 98.05 |
| JC-B-short | **97.39** | 86.37 | 85.94 |
| CJ-A-long | 96.64 | **97.15** | 21.04 |
| JC-A-long | **98.82** | 78.59 | 77.30 |
| CJ-B-long | **97.20** | 95.60 | 42.08 |
| JC-B-long | **97.21** | 75.62 | 74.51 |

Table 2: Program accuracy data for CoffeeScript-JavaScript translation tasks. Here, the tree-transformer is compared to Chen et al.'s tree-to-tree model (Chen et al. (2018)) which have previously produced state-of-the-art results. The tree-transformer has improved results on over half the datasets, and particularly shows improved performance on larger datasets.

|  | Seq2Tree Tform | Seq2Seq Tform | Literature |
|---|---|---|---|
| Jobs | 84.3 | **85.0** | **90.7** (Liang et al. (2011)) |
| Geo | **84.6** | 81.1 | **89.0** (Kwiatkowski et al. (2013)) |
| Atis | **86.4** | 84.4 | 84.6 (Dong & Lapata (2016)) |
| Ifttt, C | **52.0** | 46.0 | **89.7** (Dong & Lapata (2016)) |
| Ifttt, C+F | **48.0** | 38.0 | **78.4** (Dong & Lapata (2016)) |
| Ifttt, F1 | **68.0** | 65.7 | **73.5** (Dong & Lapata (2016)) |

Table 3: Metrics for semantic parsing tasks. The sequence-to-tree transformer outperforms the baseline transformer on most tasks. This demonstrates that the induced bias of explicit tree structure outweighs the additional hurdle of converting between positional encoding schemes. Transformer architectures in general, however, do not yet compete with state-of-the-art results.

of the art on the most difficult tasks here. Overall, these results are promising for applying tree-transformers to larger-scale tree-to-tree scenarios.

## 5.2 Sequence-to-tree: semantic parsing

For sequence-to-tree evaluation, we focused on several semantic parsing tasks to benchmark exist. These tasks present natural language queries, with the task of converting them into tree-structured code snippets for a given API. The four datasets we consider are Jobs (Califf & Mooney (1999)), a job listing database retrieval task; Geo (Tang & Mooney (2001)), a geography database retrieval task; Atis (Dahl et al. (1994)), a flight booking task; and Ifttt (Quirk et al. (2015)), a task for constructing programs with simple trigger-action forms. Our most important results are for Atis and Ifttt as they provide ample data. Jobs and Geo provide far less data, each featuring under 1000 training examples, so their results are less reliable. Nevertheless, we provide them for completeness due to their popularity in the literature.

Our metrics are chosen to properly compare our model against the literature. For Jobs, Geo, and Atis, we provide whole program accuracy as our key metric. For Ifttt, whole program accuracy is a less common metric due to the task's difficulty. Instead we report the same metrics recommended in (Quirk et al. (2015)), namely accuracy over channels; accuracy over both channels and functions; and balanced F1 score. We preprocess all datasets to match (Dong & Lapata (2016)): namely, we filter all tokens that appear less than five times in Ifttt, and we filter all source-side tokens that appear less than twice in Jobs, Geo, and Atis. For all datasets, we use a sequence-to-tree transformer with $d_{model} = 256, d_{ff} = 1024$, and $d_{pos} = 2048$.

The results for our semantic parsing experiments can be found in Table 3. Here, we compare our metrics against the best in the literature as surveyed by (Dong & Lapata (2016)). We see that on almost every metric, our model outperforms the sequence-to-sequence transformer by several percentage points. In particular, our model shows improvement on the larger datasets; it only falters on Jobs, where it performs slightly worse than the baseline transformer. It may be possible to improve our results on this smaller dataset through a cross-validated hyperparameter search, though we do

not explore that here. Enforcing hierarchical structure upon the transformer appears to be worth the additional challenge of converting between positional encodings. However, both the sequence-to-sequence and sequence-to-tree transformers perform worse than state-of-the-art recurrent methods such as Dong and Lapata's sequence-to-tree model (Dong & Lapata (2016)). This may be due to the advantages of explicitly sharing an embedding space between sequences and trees, as opposed to having to learn a positional encoding conversion scheme.

## 6 RELATED WORK

Although ours is the first effort in applying Transformer models to hierarchically shaped data, there has been a range of prior work in tree-structured extensions of recurrent architectures. Soon after the recent resurgence of recurrent neural networks over linear sequences, researchers began to consider extensions of these models that accommodate structures more complex than linear chains. Initial efforts focused on input tree structures, where the shape of the input tree is fixed in advance. Tree-LSTMs demonstrated benefits in tasks such as sentence similarity, sentiment analysis (Tai et al. (2015)), and information extraction (Miwa & Bansal (2016)). With a few changes, these models can be extended to cover graph-like structures as well (Peng et al. (2017)).

Sequence-to-sequence models without explicit tree modeling have been applied to tree generation using only a simple linearization of the tree structure (Vinyals et al. (2015); Eriguchi et al. (2017); Aharoni & Goldberg (2017)). Later work has proposed generation methods that are more sensitive to tree structures and well-formedness constraints (Dong & Lapata (2016); Alvarez-Melis & Jaakkola (2017)), leading to new-state-of-the-art results.

Rather than explicitly modeling hierarchically structured data, some recent work imposes hyperbolic geometry on the activations of neural networks (Gulcehre et al. (2018)). Defining attention in terms of hyperbolic operations allows modeling of latent hierarchical structures. In contrast, our work focuses on the case of explicit hierarchical structure.

One particularly relevant transformer variant explicitly captures relative position, rather than relying on sinusoidal models to indirectly model distances (Shaw et al. (2018)). It is a clear precursor to modeling labeled, directed graphs, though the approach detailed in the paper is limited to relative linear positions.

## 7 CONCLUSION

We have proposed a novel scheme of custom positional encodings to extend transformers to tree-domain tasks. By leveraging the strengths of the transformer, we have achieved an efficiently parallelizable model that can consider relationships between arbitrary pairs of tree nodes in a single step. Our experiments have shown that our model can often outperform sequence-transformers in tree-oriented tasks. We intend to experiment with employing the model on other tree-domain tasks of interest as future work.

By abstracting the transformer's positional encodings, we have established the potential for generalized transformers to consider other nonlinear structures, given proper implementations. As future work, we are interested in exploring alternative implementations for other domains, in particular graph-structured data as motivated by structured knowledge tasks.

Finally, in this paper we have only considered binary trees: in particular, binary tree representations of trees not originally structured as such. Arbitrary tree representations have their own advantages and complications; we would like to explore training on them directly.

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
