# OpenReview forum: "Novel positional encodings to enable tree-structured transformers"
_ICLR.cc/2019/Conference_

### Official Review · AnonReviewer3 · 2018-11-01
**An interesting tree-structured positional embedding**

**Rating:** 6
**Confidence:** 3

**Review:**

This work proposes a novel tree structure positional embedding by uniquely representing each path in a tree using a series of transformation, i.e., matmul for going up or down the edges. The tree encoding is used for transformer and shows gains over other strong baselines, e.g., RNNs, in synthetic data and a program translation task.

Pros:

- An interesting approach for representing tree structure encoding using a series of transformation. The idea of transformation without learnable parameters is novel.

- Better accuracy both on synthetic tasks and code translation tasks when compared with other strong baselines.

Cons:

- Computation seems to be larger given that the encoding has to be recomputed in every decoding step. I'd like to know the latencies incurred by the proposed method.

Other comment:

- I'd like to see experimental results on natural language tasks, e.g., syntax parsing.

- Section 2:  "we see that is is not at all necessary" -> that is

- Section 3: Notation is a little bit hard to follow, ":" for D and U, and ";" in stacking.

---

> ### Comment · Area_Chair1 · 2018-11-07
> **Please expand**
>
> Hello reviewer 3,
>
> Your review, while longer than the others on this paper, is very short. You are the only one supporting acceptance of this paper. Could you give a bit more detail about what its strengths and contributions are, and what areas it could improve on or be more clear about?
>
> Best,
> AC

---

> ### Author Response · Authors · 2018-11-27
> **Thank you for your feedback!**
>
> Hi,
>
> Thank you very much for your feedback! We have explicitly clarified potentially confusing notation in our revised draft.
>
> Regarding latency: There is no additional latency during training time as positional  encodings are directly provided by the teacher. During evaluation-time decoding, there is some extra computation needed to compute one more positional encoding per time step, but this pales in comparison to the stack of attention functions and matrix multiplications each time step demands anyway.
>
> Regards!

---

### Official Review · AnonReviewer2 · 2018-11-02
**Interesting but incremental**

**Rating:** 4
**Confidence:** 3

**Review:**

The paper describes an interesting idea for using Vashwani's transformer with tree-structured data, where nodes' positions in the tree are encoded using unique affine transformations. They test the idea in several program translation tasks, and find small-to-medium improvements in performance.

Overall the idea is promising, but the work isn't ready for publication. The implementation details weren't easy to follow, the experiments were narrow, and there are key citations missing. I would recommend trying some more diverse tasks, and putting this approach against other graph neural network techniques.


REVISED:
I've revised by review upwards by 1, though I still recommend rejection. The authors improved the scholarship by adding many more citations and related work. They also made the model details and implementation more clear.

The remaining problem I see is that the results are just not that compelling, and the experiments do not test any other graph neural network architectures.

Specifically, in Table 1 (synthetic experiments) the key result is that their tree-transformer outperforms seq-transformer on structured input. But seq-transformer is best on raw programs. I'm not sure what to make of this. But I wouldn't use tree-transformer in this problem. I'd use seq-transformer.

In Table 2 (CoffeeScript-JavaScript experiments), no seq-transformer results are presented. That seems... suspicious. Did the authors try those experiments? What were the results? I'd definitely like to see them, or an explanation of why they're not shown. This paper tests whether tree-transformers are better than seq-transformer and other seq/tree models, but this experiment's results do not address that fully. Of the 8 tasks tested, tree-transformer is best on 5/8 while tree2tree is best on 3/8.

In Table 3, there's definitely a moderate advantage to using tree-transformer over seq-transformer, but in 5/6 of the tasks tree-transformer is worse than other approaches. The authors write, "Transformer architectures in general, however, do not yet compete with state-of-the-art results.".

Finally, no other graph neural network/message-passing/graph attention architectures are tested (eg. Li et al 2016 was cited but not tested, and Gilmer et al 2017 and Veličković et al 2017 weren't cited or tested), but there's a reasonable chance they'd outperform the tree-transformer.

So overall the results are intriguing, and I believe there's something potentially valuable here. But I'm not sure there's sufficient reason presented in the paper to use tree-transformer over seq-transformer or other seq/tree models. Also, while the basic idea is nice, as I understand it is restricted to trees, so other graphical structures wouldn't be handled.

---

> ### Comment · Area_Chair1 · 2018-11-07
> **More detail needed**
>
> Hello Reviewer 2,
>
> Thank you for your review, but I'm afraid a little more detail is needed to justify your score, as your review is quite short.
>
> In particular, what about the experiments makes them too narrow? What additional experiments would you like to see? What key citations are needed? In particular, what graph neural network approaches would you recommend comparing against?
>
> It is essential, when recommending rejection, that a coherent argument be made for it so that the authors have something to respond to or rebut, or at least critical feedback they can use when revising the paper.
>
> Best,
> AC

---

> ### Author Response · Authors · 2018-11-27
> **Thank you for your feedback!**
>
> Hi,
>
> Thank you kindly for your feedback. In our revision we have made an effort to clarify implementation details, add more results from experiments, and expand our citations.
>
> Regards!

---

### Official Review · AnonReviewer1 · 2018-11-05
**Promising approach for enabling transformers to process tree-structured data**

**Rating:** 5
**Confidence:** 3

**Review:**

The authors propose to change the positional encodings in the transformer model to allow processing of tree-structured data.
The tree positional encodings summarize the path between 2 nodes as a series of steps up or down along tree branches with the constraint that traveling up a branch negates traveling down any branch.

The experimental results are encouraging and the method notably outperforms the regular transformer as well as the tree2tree LSTM introduced by Chen et al on larger datasets.

The current draft lacks some clarity and is low on references. It would also be interesting to see experiments with arbitrary trees or at least regular trees with degree > 2 (rather than just binary trees). While the authors only consider binary trees in this paper, it represents a good first step towards generalizing attention-based models to nonlinear structures.

Comments:
* Would it be possible to use the fact that D_kU = I for the correct branch k? (This happens frequently for binary trees)

---

> ### Comment · Area_Chair1 · 2018-11-07
> **More detail needed**
>
> Hello reviewer 1,
>
> Thank you for your review, but I'm afraid a little more detail is needed to justify your score, as your review is quite short.
>
> In particular, what are some references that you feel are missing? Why is it crucial to show experiments with larger trees, since, for example, any grammar can be binarised by putting it into Chomsky Normal Form? In what areas is the paper unclear, and could this be rectified during the rebuttal period?
>
> Thanks.
> AC

---

> > ### Comment · AnonReviewer1 · 2018-11-09
> > **More details**
> >
> > The current draft has no related work section and does not put the research in context with the existing literature. It contains only a mere 7 references, 3 of which are Transformer (the model used), Adam (the optimizer used) and Chen et al (the only baseline).
> > It ignores the many works that have used position embeddings / encodings before such as [1, 2, 3]. I also suspect that there exist spectral theory approaches to represent nodes in a tree (consider the eigenvectors of the adjacency matrix for example)
> >
> > Regarding clarity: Some notation is not introduced in section 3. Dimensions are not always obvious and more figures would help with comprehension.
> >
> > Concerning binary trees: There are a few non-discussed issues about relying on the Left-Child Right-Sibling Representation, such as whether any information is lost (this changes the number of branches between nodes) and how the increase of the tree depth affects downstream performance (since the encodings can only encode information perfectly up to k branches). The trade-off between n and k is also not discussed (for example when does it become useful to represent a n-ary tree into its Left-Child Right-Sibling Representation?).
> >
> > [1] Self-Attention with Relative Position Representations
> > [2] Music Transformer
> > [3] Convolutional Sequence to Sequence Learning

---

> > > ### Author Response · Authors · 2018-11-27
> > > **Thank you for your feedback!**
> > >
> > > Hi,
> > >
> > > Thank you very much for your feedback.
> > >
> > > In our revised draft, we have done our best to address your concerns. We have added a related work section to better ground our contribution, and we have tried to clarify sections 3 and 4 with some additional detail and figures. We appreciate you pointing out the clarity issues and your recommendations on relevant literature to cite.
> > >
> > > In regards to spectral theory approaches to tree node representation, that is a very interesting idea with a lot of promise. It would however be difficult to directly implement within our paradigm. In our system, and in transformers in general, it is assumed that the values of the decoder inputs and positions do not change over time. But as we build up a tree over multiple time steps, its adjacency matrix and associated eigenvectors change. This hinders us from directly using these eigenvectors as positional encodings.
> > >
> > > In regards to binary trees: binary tree representations have been used extensively in NLP literature, and the LCRS representation in particular allows us to directly compare our work with other recent program translation literature. One key benefit to binary tree representations is that they let us work with trees with widely varying degrees among nodes, e.g. abstract syntax trees featuring functions that take arbitrary numbers of arguments. We do agree that binary tree representations have some issues, and are interested in exploring k-ary trees in future work.
> > >
> > > Regards!

---

### Meta-Review · Area_Chair1 · 2018-12-14
**Borderline and not ideally reviewed, but not quite ready**

**Confidence:** 2
**Recommendation:** Reject

**Metareview:**

This paper extends the transformer model of Vashwani et al. by replacing the sine/cosine positional encodings with information reflecting the tree stucture of appropriately parsed data. According to the reviews, the paper, while interesting, does not make the cut. My concern here is that the quality of the reviews, in particular those of reviewers 2 and 3, is very sub par. They lack detail (or, in the case of R2, did so until 05 Dec(!!)), and the reviewers did not engage much (or at all) in the subsequent discussion period despite repeated reminders. Infuriatingly, this puts a lot of work squarely in the lap of the AC: if the review process fails the authors, I cannot make a decision on the basis of shoddy reviews and inexistent discussion! Clearly, as this is not the fault of the authors, the best I can offer is to properly read through the paper and reviews, and attempt to make a fair assessment.

Having done so, I conclude that while interesting, I agree with the sentiment expressed in the reviews that the paper is very incremental. In particular, the points of comparison are quite limited and it would have been good to see a more thorough comparison across a wider range of tasks with some more contemporary baselines. Papers like Melis et al. 2017 have shown us that an endemic issue throughout language modelling (and certainly also other evaluation areas) is that complex model improvements are offered without comparison against properly tuned baselines and benchmarks, failing to offer assurances that the baselines would not match performance of the proposed model with proper regularisation. As some of the reviewers, the scope of comparison to prior art in this paper is extremely limited, as is the bibliography, which opens up this concern I've just outlined that it's difficult to take the results with the confidence they require. In short, my assessment, on the basis of reading the paper and reviews, is that the main failing of this paper is the lack of breadth and depth of evaluation, not that it is incremental (as many good ideas are). I'm afraid this paper is not ready for publication at this time, and am sorry the authors will have had a sub-par review process, but I believe it's in the best interest of this work to encourage the authors to further evaluate their approach before publishing it in conference proceedings.